# Initial waterline contamination by *Pseudomonas aeruginosa* in newly installed dental chairs

Alexandre Baudet,[1,2,3] Julie Lizon,[4] Arnaud Florentin,[3,4,5] Éric Mortier[1,2,6]

**ABSTRACT** Water contamination in dental unit waterlines (DUWLs) is a potential source of healthcare-associated infection during dental care. The aim of this study was to evaluate the microbiological quality of DUWLs water from newly installed dental chairs in a French University Hospital. The microbiological quality of water from 24 new DUWLs initially disinfected by ICX Renew—prior to use of the dental units for patient treatment—was assessed for total culturable aerobic bacteria at 22°C and 36°C, *Legionella* sp., *Pseudomonas aeruginosa*, and total coliforms. Among the 24 samples analyzed, 21 were compliant with the water quality levels: 19 had no bacteria, and 2 contained only 4 and 1 CFU/mL for total culturable aerobic bacteria at 22°C and 36°C, respectively. Three samples were non-compliant due to contamination by *P. aeruginosa* (4, 2, and 2 CFU/100 mL). Controlling and preventing the microbiological contamination of DUWLs, especially by pathogenic bacteria, at the time of the installation of the new dental chairs are crucial to prevent healthcare-associated infection in dentistry.

**IMPORTANCE** Dental unit waterlines (DUWLs) of new dental chairs may be contaminated before their first clinical use, so an initial shock disinfection is crucial at the time of their installation. The microbiological analyses are crucial to control the water quality of DUWLs before their first clinical use because their disinfection does not guarantee the elimination of all bacteria.

**KEYWORDS** water quality, infectious control, dental chair, waterlines, water microbiology

The water emerging from the waterlines of dental chairs—also called dental unit waterlines (DUWLs)—may expose patients by contact (by skin, mucosa, bone, or vascular contact during dental care), by ingestion, and also by inhalation (due to aerosol-generating procedures with the high-speed dental turbines and handpieces, the air/water syringe, and the ultrasonic scalers). These water expositions may potentially lead to healthcare-associated infections (1). Indeed, dental care with contaminated DUWLs water has generated infections with *Mycobacterium abscessus* among many children (2–4), dental abscesses due to *Pseudomonas aeruginosa* among immunocompromised patients (5), and deaths caused by *Legionella pneumophila* (6, 7).

The outlet water of DUWLs is well known to often present a poor microbiological quality compared to the inlet water (1) due to the retrograde contamination by oral fluids (8–10) and to the formation of a biofilm within the DUWLs promoted by their strong complexity, by their low water flow, and by the water stagnation during inactivity periods (11–13).

To date, the water contamination of DUWLs has been largely studied among dental chairs already used for several years (1). But, to our knowledge, no study has investigated the possible contamination of DUWLs among new dental chairs before their first clinical use. We hypothesized that the waterlines of newly installed dental chairs may

Address correspondence to Alexandre Baudet, alexandre.baudet@univ-lorraine.fr.

The authors declare no conflict of interest.

be contaminated before their installation in relation with the quality tests performed by the manufacturer during the manufacturing process of the dental chairs. The aim of this study was to evaluate the microbiological quality of DUWLs water from 24 newly installed dental chairs in the University Hospital of Nancy, France.

## MATERIALS AND METHODS

### Dental chairs

At the end of 2022, the Dental Department of the University Hospital of Nancy in France acquired 24 new dental chairs manufactured by A-Dec (A-dec, Inc., Oregon, USA). The dental chairs were manufactured in the United States of America (USA) from April to May 2022, apart from two manufactured in September (Fig. 1). Next, they were delivered in France, in June, by boat within 7 days (except for two dental chairs manufactured and delivered in September). Then, they were delivered by truck and stored at the provider until their installation at the end of November. The dental chairs were installed in the same hospital building on two floors: 12 on the first floor and 12 on the second floor.

The dental chairs were not equipped with a water cup filler. All of them were equipped with an independent tank—a bottle—to supply the DUWLs with sterile water and disinfectants. During their installation, on 25 and 28 November, a leak test of each DUWL was performed by the fitter using the hospital's main water supply. Then the DUWLs were initially shock disinfected, and supplied with disinfectants and sterile water. Finally, the water was microbiologically analyzed.

### Disinfection of the dental unit waterlines

On 30 November, an initial shock disinfection was performed by the fitter to remove the potential bacteria and biofilms inside all DUWLs with ICX Renew (A-dec, Inc., Oregon, USA) according to the manufacturer's instruction. ICX Renew consists of hydrogen peroxide, sodium lauryl sulfate, and maleic acid.

According to our hospital procedures developed since 2013 and described in a previous study (14), Alpron and Bilpron disinfectants (Alpro Medical GmbH, Germany) were continuously used in the DUWLs. Alpron is used during periods of activity (Monday to Friday) while Bilpron is used during inactivity periods (weekends). So, on 2 December (Friday), the ICX Renew was flushed and replaced by:

- Alpron diluted at 1% concentration in the water supply bottle with sterile water for the DUWLs on the second floor. Alpron at 1% is a bacteriostatic disinfectant for the continuous maintenance of the water quality in DUWLs. It is mainly made up of ethylene diamine tetra acetic acid (EDTA), polyhexamethylene biguanide, and tosylchloramide sodium.
- Bilpron is a solution ready for use, which was used unmixed for the DUWLs on the first floor because the laboratory team did not have enough time to collect and culture the DUWLs water samples before the weekend. Bilpron is a bactericidal disinfectant used during the inactivity periods exceeding 24 hours of DUWLs. Bilpron contains EDTA, polyhexamethylene biguanide, and ester para-hydroxybenzoate. On 5 December, the Bilpron was flushed and replaced by Alpron at 1% with sterile water.

### Sampling

Sampling was performed on 2 and 6 December, respectively, on the second and first floors. Water samples (500 mL) were taken simultaneously from the water syringes and from the micromotor and handpiece supply tubes. Each of the six DUWLs' outlets of the dental chairs (two water syringes, two micromotor supply tubes, one turbine handpiece supply tube, and one ultrasonic handpiece supply tube) counted for 1/6 of the total water sample. Water samples were collected aseptically in sterile containers with a

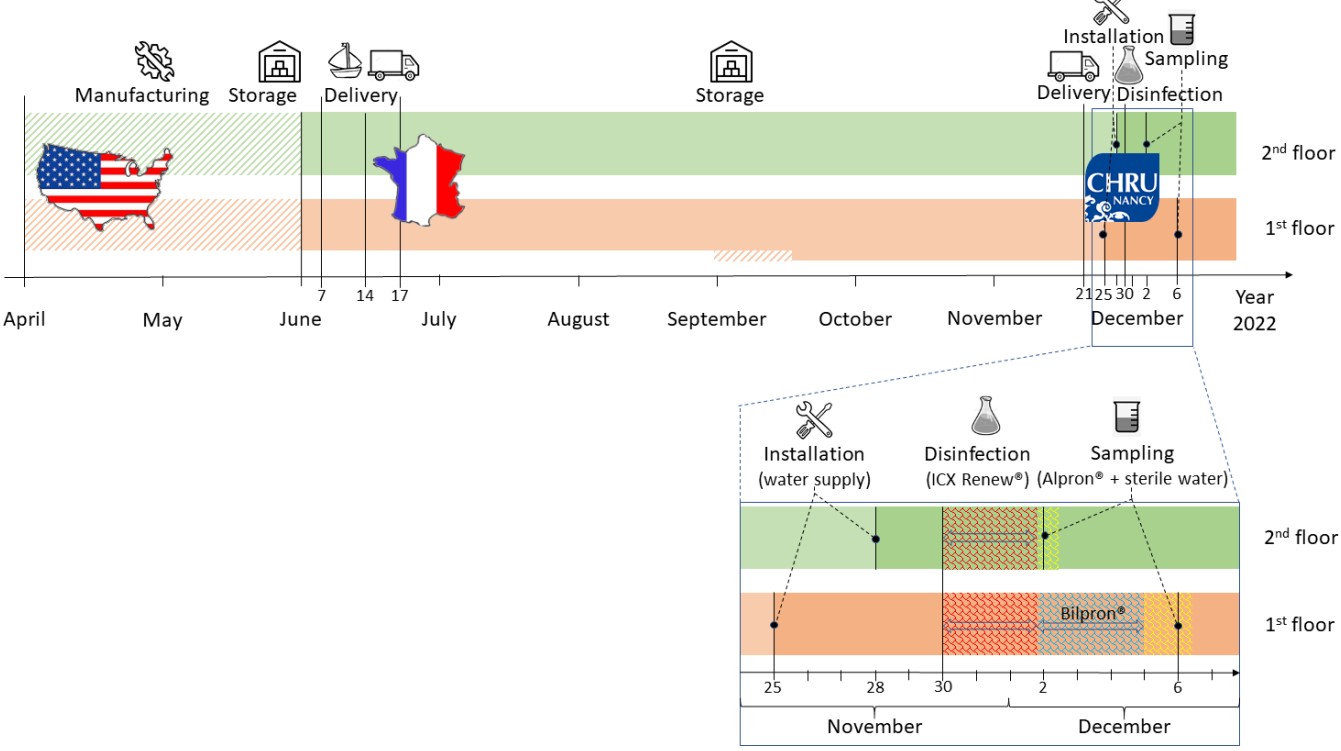

FIG 1 Dental chair history.

filtered mixture (20 mL) of sodium thiosulfate, Tween 80, lecithin, and histidine to inhibit the Alpron disinfectant diluted at 1%. Samples were transferred to the laboratory within 2 hours in insulated boxes and quickly processed in the following way.

## Microbiological analysis

Each water sample was subjected to analysis for total culturable aerobic bacteria (TCAB) at 22°C and 36°C, *Legionella* sp., *Pseudomonas aeruginosa*, and total coliform bacteria including *Escherichia coli*, in the Laboratory of Environmental Biology of the University Hospital.

The details of the microbiological analysis were described in our previous study (14). The detection of the bacteria was performed from each water sample of 500 mL according to the international standards for water quality: NF EN ISO 6222 for TCAB (15) using $2 \times 1$ mL of sampled water, NF EN ISO 11731-2 for *Legionella* sp. and *L. pneumophila* (16) using 0.2, 10, and 100 mL of sampled water, NF EN ISO 16266 for *P. aeruginosa* (17) using 100 mL of sampled water, NF EN ISO 9308-1 for *Escherichia coli* and coliform bacteria (18) using 100 mL of sampled water.

The microbiological water quality levels were determined in a previous publication (19) according to the French guidelines about water in healthcare facilities (20). These levels are presented in Table 1.

## Corrective measure and control sampling

After the first sampling, all the DUWLs were flushed to replace Alpron at 1% by Bilpron. On 19 December, the DUWLs presenting non-compliant microbiological results were treated by BRS (Alpro Medical GmbH, Germany). BRS was used because our hospital procedures recommend the use of BRS when a microbiological analysis of DUWLs water is non-compliant (action level) based on a previous study carried out in our hospital (14). BRS is a product for shock disinfection used to remove the bacteria and biofilms inside DUWLs. It consists of a two-phase basic cleaning system to be put in the DUWLs in this

**TABLE 1** Interpretation of the results of the microbiological analysis of DUWLs water

| Results | Water quality levels |
|---|---|
| TCAB at 36°C ≤10 CFU/mL and | |
| TCAB at 22°C ≤100 CFU/mL and | Compliant (target level) |
| Absence of pathogens (*L. pneumophila, P. aeruginosa*, etc.) | |
| TCAB at 36°C >10 and ≤100 CFU/mL or | |
| TCAB at 22°C >100 and ≤300 CFU/mL and | |
| Absence of pathogens (*L. pneumophila, P. aeruginosa*, etc.) | Acceptable (alert level) |
| TCAB at 36°C >100 CFU/mL or | |
| TCAB at 22°C >300 CFU/mL or | Non-compliant (action level) |
| Presence of pathogens (*L. pneumophila, P. aeruginosa*, etc.) | |

order: BRS PreCleaner (an enzymatic cleaning agent containing sodium carbonate and disodium metasilicate), then mixed solutions of BRS Remover (citric acid crystals) and BRS Activator (surfactants), followed by rinsing with sterile water and disinfection with Bilpron for 12 hours contact time. Finally, the Bilpron was flushed and replaced by Alpron at 1% with sterile water for a control sampling on 20 December.

## Tap water

Six water samples were collected on 21 December from taps in six different rooms where dental chairs were installed. The hospital's main water samples from taps were analyzed for TCAB at 22°C and 36°C, *P. aeruginosa* and total coliform bacteria.

## Statistical analysis

Microbiological data were collected in Microsoft Excel (Microsoft Corporation, Washington, USA). They were described as numbers and percentages.

## RESULTS

The microbiological analysis of the 24 water samples revealed no coliform and no *Legionella* sp., and 21 samples (87.5%) were compliant with the water quality levels.

Among the 12 water samples on the first floor (ICX Renew + Bilpron), all of them (100%) were compliant without any bacteria detected (Table 2).

Among the 12 water samples on the second floor (ICX Renew), three (25%) were non-compliant because of *P. aeruginosa* contamination: 4, 2, and 2 CFU/100 mL (Table 3). The other nine samples (75%) were compliant. One sample presented 1 CFU/mL for TCAB at 36°C and two samples presented 4 CFU/mL for TCAB at 22°C including one non-compliant sample.

After the BRS disinfection of the three non-compliant DUWLs, all the control samples were compliant without any bacteria detected.

Regarding the six water samples (250 mL) collected from taps, no *P. aeruginosa* and no coliform were detected. For TCAB at 36°C, no bacterium was detected, except 1 CFU/mL in one sample. For TCAB at 22°C, 0, 1, 1, 2, 3, and 5 CFU/mL were counted in the six samples, respectively.

## DISCUSSION

This study highlights that the DUWLs of new dental chairs before their first clinical use can be contaminated by bacteria, and more specifically by *P. aeruginosa* despite an initial shock disinfection. *P. aeruginosa* is a pathogenic and waterborne bacterium frequently found in DUWLs outlet water (21–24); it can also sometimes contaminate surfaces in dental clinics (23, 25). This bacterium colonizing and forming biofilms inside DUWLs is associated with several diseases in immunocompromised people (26, 27). It is an opportunistic pathogen associated with many types of infection

TABLE 2  Microbiological analysis of the DUWLs treated by ICX Renew + Bilpron on the first floor

| Dental chair | TCAB at 22°C (CFU/mL) | TCAB at 36°C (CFU/mL) | *P. aeruginosa* (CFU/100 mL) | Coliforms (CFU/100 mL) | *Legionella* sp. (CFU/100 mL) | Water quality levels |
|---|---|---|---|---|---|---|
| 1 | 0 | 0 | 0 | 0 | <10 | Compliant |
| 2 | 0 | 0 | 0 | 0 | <10 | Compliant |
| 3 | 0 | 0 | 0 | 0 | <10 | Compliant |
| 4 | 0 | 0 | 0 | 0 | <10 | Compliant |
| 5 | 0 | 0 | 0 | 0 | <10 | Compliant |
| 6 | 0 | 0 | 0 | 0 | <10 | Compliant |
| 7 | 0 | 0 | 0 | 0 | <10 | Compliant |
| 8 | 0 | 0 | 0 | 0 | <10 | Compliant |
| 9 | 0 | 0 | 0 | 0 | <10 | Compliant |
| 10 | 0 | 0 | 0 | 0 | <10 | Compliant |
| 11 | 0 | 0 | 0 | 0 | <10 | Compliant |
| 12 | 0 | 0 | 0 | 0 | <10 | Compliant |

including hospital-acquired pneumonia, skin infections, urinary tract infections, burns, eye infections, and bloodstream infections (28).

Regarding new DUWLs, the origin of the bacterial contamination cannot come from retrograde contamination because the dental chairs had never been used with patients. So, the contamination of the new DUWLs comes from the water which supplied the dental chairs: the supply water is recognized as the main source of DUWLs contamination (29). In our study, this contamination could not have originated from the water supply after the initial shock disinfection of the DUWLs because the dental chairs were supplied by an independent reservoir bottle filled with sterile water and disinfectants. We can formulate three hypotheses to explain this initial contamination of the new DUWLs: (i) the contamination occurred during the manufacturing of the dental chairs which include quality tests probably performed with the main water of the factory, then the bacteria have grown and formed biofilms during their storage for several months and their deliveries (by boat and trucks); (ii) the contamination occurred during the installation of the dental chairs due to the presence of *P. aeruginosa* present on tools/equipment used by the fitter; (iii) the contamination originated from the leak test performed by the fitter with main water of the hospital at the installation time, then the bacteria have grown and formed biofilms for 2–5 days before their initial shock disinfection. The first hypothesis is more credible because a long period of inactivity (deliveries and storage) with stagnant water originated from the factory may have facilitated the growth and formation of biofilms inside the DUWLs. However, no samples were collected before the dental chair installation to verify this hypothesis and no

TABLE 3  Microbiological analysis of the DUWLs treated by ICX Renew on the second floor

| Dental chair | TCAB at 22°C (CFU/mL) | TCAB at 36°C (CFU/mL) | *P. aeruginosa* (CFU/100 mL) | Coliforms (CFU/100 mL) | *Legionella* sp. (CFU/100 mL) | Water quality levels |
|---|---|---|---|---|---|---|
| 13 | 0 | 0 | 0 | 0 | <10 | Compliant |
| 14 | 0 | 0 | 0 | 0 | <10 | Compliant |
| 15 | 0 | 0 | 0 | 0 | <10 | Compliant |
| 16 | 0 | 0 | 4 | 0 | <10 | Non-compliant |
| 17 | 0 | 0 | 2 | 0 | <10 | Non-compliant |
| 18 | 0 | 0 | 0 | 0 | <10 | Compliant |
| 19 | 4 | 0 | 0 | 0 | <10 | Compliant |
| 20 | 0 | 0 | 0 | 0 | <10 | Compliant |
| 21 | 0 | 0 | 0 | 0 | <10 | Compliant |
| 22 | 0 | 0 | 0 | 0 | <10 | Compliant |
| 23 | 0 | 1 | 0 | 0 | <10 | Compliant |
| 24 | 4 | 0 | 2 | 0 | <10 | Non-compliant |

previous article explored these findings. Future studies will be needed to confirm or refute this hypothesis. The second hypothesis is less credible because a cross-contamination from tools/equipment to the inner surfaces of DUWLs seems complex, and only 2–5 days separated the installation of the DUWLs of their initial shock disinfection. The colonization and the formation of biofilms during the earliest stage of bacteria adhesion over this short period should probably be neutralized by the initial shock disinfection by ICX Renew. The third hypothesis can be rejected because six water samples were collected from different taps of the two floors in the building and these samples from the hospital's main water revealed no *P. aeruginosa*.

To our knowledge, the efficiency of the initial shock disinfection of DUWLs by ICX Renew which containing hydrogen peroxide was never studied. Only the continuous disinfection of DUWLs with ICX was tested (30, 31). In a previous study, Abdouchakour et al. showed that some colony-forming units of *P. aeruginosa* could survive in a few DUWLs despite a shock disinfection with hydrogen peroxide. The resistance of *P. aeruginosa* strains to several biocides has been confirmed by antimicrobial susceptibility tests (21). This study highlights the use of a single disinfectant such as ICX Renew for the initial shock disinfection of DUWLs (second floor) seems less effective than if it is combined with a second disinfectant such as Bilpron (first floor).

Regarding the three dental chairs (numbers 16, 17, and 24) contaminated on the second floor after the initial shock disinfection with ICX Renew, we noticed that analysis of dental chairs 16 and 17 revealed no TCAB (0 CFU/mL) while *P. aeruginosa* strains were found (4 and 2 CFU/100 mL). The detection of *P. aeruginosa* without TCAB can be explained by the difference in the volume of water analyzed: 1 mL for TCAB at 22°C and 1 mL for TCAB at 36°C compared to 100 mL for *P. aeruginosa*. To treat these three dental chairs, we used BRS for a second shock disinfection according to our hospital procedures because it contains different biocides and enzymatic cleaning agents which reduce the risk of resistance, and its efficiency was previously demonstrated to decontaminated DUWLs (14). The use of BRS also permitted to obtain compliant results in this study. So, BRS appears as a pragmatic solution that dentists and infection prevention and control teams could use to perform a shock disinfection when DUWLs are contaminated by bacteria and biofilms.

To the authors' knowledge, this study is the first to analyze the microbiological water quality of DUWLs in newly installed dental chairs before their first clinical use. In the literature of the last 10 years, different DUWL disinfectants were tested in experimental conditions (31–36) and in routine practice with dental chairs already used for patient treatment (19, 37–46). It was shown that a shock disinfection before continuous disinfection offers a more rapid effect to reduce the quantity of bacteria in DUWLs (37). Moreover, in routine practice, a combined protocol of continuous and periodic disinfection with different active products was more efficient than solely continuous or solely periodic disinfection to control bacterial contamination of the DUWLs water (19, 38).

To date, there is no standard about quality control of DUWL during the production of dental chairs. In dental offices, the water quality monitoring by sample analysis is recommended in the USA in order to avoid massive microbiological contamination and to detect the presence of pathogenic bacteria (47–50). In France, less than 3% of dentists perform microbiological analysis of their DUWLs water (51). A first microbiological analysis—at the installation time—is not performed by dental unit manufacturers or providers to prove the safety of their dental chairs which are medical devices. Moreover, dental unit manufacturers do not use disinfectants to prevent biofilm growth before the medical device installation. This study highlights the need to perform microbiological analysis of DUWLs water routinely but also before their first clinical planned use. Indeed, despite an initial shock disinfection performed according to the manufacturer's instructions, 12.5% of DUWLs appeared contaminated by *P. aeruginosa*.

A limitation of this study is the absence of microbiological analysis before the initial shock disinfection to evaluate the efficiency of the disinfectants used. The quantity

of bacteria detected after the initial shock disinfection is poor, but the reduction rate offered by the disinfectants used cannot be evaluated. Further studies are required to assess the contamination rate of new DUWLs before the initial shock disinfection.

*P. aeruginosa* is one of the most important and difficult to eliminate pathogens commonly found in DUWLs. *P. aeruginosa* originated from DUWLs may generate healthcare-associated infections like dental abscesses among immunocompromised patients (5). Case reports showed that dental infections by *P. aeruginosa* could cause brain abscesses (52), epidural abscesses, and cervical osteomyelitis (53). Controlling and preventing the microbial contamination of DUWLs water are crucial to prevent healthcare-associated infection. In order to control the water contamination of DUWLs, an internal control plan is necessary, including an initial shock disinfection followed by a microbiological analysis to control its efficiency. The water quality monitoring of DUWLs is useful not only to assess the efficiency of preventive measures but also to guide the implementation of corrective strategies.

Further studies with microbiological analysis of DUWLs water before the leak test at the installation time of new dental chairs should be performed to better identify the contamination origin and to evaluate the efficacy of the initial shock disinfection procedures recommended by the manufacturers.

## Conclusions

An initial shock disinfection of DUWLs by ICX Renew performed according to the manufacturer's instructions appeared efficient to control the water quality regarding TCAB. However, 12.5% of DUWLs remained contaminated by *P. aeruginosa* despite this initial shock disinfection. Microbiological analysis of DUWLs water should be performed routinely but also before their first clinical planned use to control the microbial quality of water used for dental care and to prevent healthcare-associated infection.

### ACKNOWLEDGMENTS

The authors are grateful to the Laboratory of Environmental Biology (CHRU Nancy) for the microbiological analyses.

The authors received no funding for this work.

Conceptualization: A.B., A.F., and E.M.; methodology: A.B., A.F., and E.M.; formal analysis: A.B. and E.M.; investigation: A.B., J.L., and E.M.; data curation: E.M. and J.L.; writing—original draft preparation: A.B. and E.M.; writing—review and editing: A.B., A.F., E.M., and J.L.; visualization: A.B.; supervision and project administration: E.M.

### AUTHOR AFFILIATIONS

[1]Faculté d'odontologie, Université de Lorraine, Nancy, France
[2]CHRU-Nancy, Service d'odontologie, Nancy, France
[3]Université de Lorraine, Inserm, INSPIIRE, Nancy, France
[4]Département territorial d'hygiène et prévention du risque infectieux (DTPRI), CHRU-Nancy, Nancy, France
[5]Département d'hygiène, des risques environnementaux et associés aux soins (DHREAS), Faculté de Médecine, Université de Lorraine, Nancy, France
[6]CNRS, IJL, Université de Lorraine, Nancy, France

### AUTHOR ORCIDs

Alexandre Baudet  http://orcid.org/0000-0003-2905-0530
Julie Lizon  http://orcid.org/0000-0001-8831-5629
Arnaud Florentin  http://orcid.org/0000-0003-2011-4016
Éric Mortier  http://orcid.org/0000-0003-3144-5929

## DATA AVAILABILITY

Data fully available in article.

## ADDITIONAL FILES

The following material is available online.

## Open Peer Review

**PEER REVIEW HISTORY (review-history.pdf).** An accounting of the reviewer comments and feedback.

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
