## [Reviewer comments · Microbiology Spectrum]

Microbiology Spectrum

Initial waterline contamination by *Pseudomonas aeruginosa* in newly installed dental chairs

Alexandre Baudet, Julie Lizon, Arnaud Florentin, and Éric Mortier

Corresponding Author(s): Alexandre Baudet, CHRU Nancy

Review Timeline:

Submission Date:	November 16, 2023
Editorial Decision:	March 1, 2024
Revision Received:	March 15, 2024
Accepted:	April 8, 2024

Editor: Erik Hom

Reviewer(s): Disclosure of reviewer identity is with reference to reviewer comments included in decision letter(s). The following individuals involved in review of your submission have agreed to reveal their identity: Xuefen Yu (Reviewer #2)

Transaction Report:

DOI: <https://doi.org/10.1128/spectrum.03962-23>

Re: Spectrum03962-23 (Initial waterline contamination by *Pseudomonas aeruginosa* in newly installed dental chairs)

Dear Dr. Alexandre Baudet:

Thank you for the privilege of reviewing your work. Below you will find my comments, instructions from the Spectrum editorial office, and the reviewer comments.

I am requesting that you revise your manuscript per the reviewers' comments. Note that both reviewers have commented that there are several points/sentences in your manuscript that are vague; please make sure to carefully go over the statements you make and ensure that they are specific and substantive. In your cover letter accompanying your revisions, please summarize the changes you have made in response. Your revised manuscript may be sent out for re-review.

Revision Guidelines

Sincerely,
Erik Hom
Editor
Microbiology Spectrum

Reviewer #1 (Comments for the Author):

This study was of great importance which proved the contamination had existed before first dental treatment. Also the study told

us the initial shock treatment should not be ignored, although the dental unit was installed first time.

I'm very glad to read this paper, because it makes me learn more about the dental unit waterlines. And I have some questions needed to discuss with the author.

1. What's the concentration of Bilpron?
2. Is the function of BRS removing the biofilms? What's the difference between ICX and BRS ? Why did you choose the BRS disinfectant? And I have not found the results of Microbiological analysis after the BRS disinfection treatment.
3. There were 24 water samples from the water syringes, micromotor and handpiece supply tubes. How many water samples from the water syringes? How many water samples from the micromotor and handpiece supply? Please list.
4. In the discussion, the second hypothesis and the third hypothesis can not be departed clearly. Because on 25 and 28 November, a leak test of each DUWL was performed by the fitter using the hospital main water supply during the installation. And the initial shock disinfection was performed on 30 November. If the contamination occurred during the installation of the dental chairs, and the microorganisms could also grow and form biofilms inside the DUWLs during 29 to 30 in November. The tube of waterline was already wet because of the leak test. Also no *P. aeruginosa* was tested from the tap water samples, which could not reveal there was no *P. aeruginosa* in tap water. Because *P. aeruginosa* could grow and form biofilms in the tube, so that they could be tested in the dental water samples and not be tested in the tap water samples.
5. I suggested that the microbiological analysis could be performed before the leak test to analyze whether the contamination origin was from the factory.

Reviewer #2 (Comments for the Author):

I have a few questions and comments to share with them:

Experimental procedures :

1. Disinfection of the Dental Unit Waterlines : why are so many different disinfectants used for disinfection prior to sampling and what is the significance of Alpron® and Bilpron® being used at this stage?
2. Sampling: Is it sufficient to use 20ml of neutralizer on a 500ml sample to remove the disinfectant? Please indicate whether the timing of the tap water collection of the sample is consistent with the timing of the dental chair waterlines collection.
3. Microbiological analysis :
 - 3.1 Please describe in detail the process of microbial culture, e.g., what temperature is taken for how many days of culture, etc., which is very unfriendly to the reader, although reference standards have been listed.
 - 3.2 "Each water sample was subjected to analysis for total culturable aerobic bacteria (TCAB) at 22 and 36{degree sign}C, *Legionella* sp., *Pseudomonas aeruginosa* and total coliform bacteria including *Escherichia coli*". This sentence is a bit vague. Please indicate whether these different culture assays use the same 500ml sample or different samples collected at the same time.
4. What is the significance of using BRS® to sterilize substandard dental chairs in the study ?

Results :

1. The results of incubation of tap water samples should also be presented in the results section.
2. In Table 3, the TCAB results for dental chairs 16 and 17 are all 0, but the results of their *P. aeruginosa* cultures are 4 and 2, respectively. The authors should try to analyze and explain this result in the Discussion section.

Discussion:

The discussion section should provide meaningful insights and offer a comprehensive comparison of the findings with relevant studies in the existing literature. This would enhance the interpretation of the results by providing a broader context and a deeper understanding of their significance. The overall logic of the discussion section is poor.

This study was of great importance which proved the contamination had existed before first dental treatment. Also the study told us the initial shock treatment should not be ignored, although the dental unit was installed first time.

I'm very glad to read this paper, because it makes me learn more about the dental unit waterlines. And I have some questions needed to discuss with the author.

1. What's the concentration of Bilpron?
2. Is the function of BRS removing the biofilms? What's the difference between ICX and BRS? Why did you choose the BRS disinfectant? And I have not found the results of Microbiological analysis after the BRS disinfection treatment.
3. There were 24 water samples from the water syringes, micromotor and handpiece supply tubes. How many water samples from the water syringes? How many water samples from the micromotor and handpiece supply? Please list.
4. In the discussion, the second hypothesis and the third hypothesis can not be departed clearly. Because on 25 and 28 November, a leak test of each DUWL was performed by the fitter using the hospital main water supply during the installation. And the initial shock disinfection was performed on 30 November. If the contamination occurred during the installation of the dental chairs, and the microorganisms could also grow and form biofilms inside the DUWLs during 29 to 30 in November. The tube of waterline was already wet because of the leak test. Also no *P. aeruginosa* was tested from the tap water samples, which could not reveal there was no *P. aeruginosa* in tap water. Because *P. aeruginosa* could grow and form biofilms in the tube, so that they could be tested in the dental water samples and not be tested in the tap water samples.
5. I suggested that the microbiological analysis could be performed before the leak test to analyze whether the contamination origin was from the factory.

This is a study on water contamination of new dental chairs before their first clinical use, who has some clinical significance for the safety of patients and healthcare professionals in dental practice. Because most of the published articles on microbiological testing of waterlines of dental chairs focus on chairs that are already in use, and it is still not clear whether the waterlines of newly installed chairs is colonized by microorganisms or not, but, as the authors mentioned in their limitations, the lack of sampling prior to the first shock disinfection is a greater pity.

I have a few questions and comments to share with them:

Experimental procedures:

1. Disinfection of the Dental Unit Waterlines: why are so many different disinfectants used for disinfection prior to sampling and what is the significance of Alpron® and Bilpron® being used at this stage?
2. Sampling: Is it sufficient to use 20ml of neutralizer on a 500ml sample to remove the disinfectant? Please indicate whether the timing of the tap water collection of the sample is consistent with the timing of the dental chair waterlines collection.
3. Microbiological analysis:
 - 3.1 Please describe in detail the process of microbial culture, e.g., what temperature is taken for how many days of culture, etc., which is very unfriendly to the reader, although reference standards have been listed.
 - 3.2 “Each water sample was subjected to analysis for total culturable aerobic bacteria (TCAB) at 22 and 36°C, *Legionella sp.*, *Pseudomonas aeruginosa* and total coliform bacteria including *Escherichia coli*”. This sentence is a bit vague. Please indicate whether these different culture assays use the same 500ml sample or different samples collected at the same time.
4. What is the significance of using BRS® to sterilize substandard dental chairs in the study?

Results:

1. The results of incubation of tap water samples should also be presented in the results section.
2. In Table 3, the TCAB results for dental chairs 16 and 17 are all 0, but the results of their *P. aeruginosa* cultures are 4 and 2, respectively. The authors should try to analyze and explain this result in the Discussion section.

Discussion:

The discussion section should provide meaningful insights and offer a comprehensive comparison of the findings with relevant studies in the existing literature. This would enhance the interpretation of the results by providing a broader context and a deeper understanding of their significance. The overall logic of the discussion section is poor.

Response to Reviewers

Reviewer #1:

This study was of great importance which proved the contamination had existed before first dental treatment. Also the study told us the initial shock treatment should not be ignored, although the dental unit was installed first time.

I'm very glad to read this paper, because it makes me learn more about the dental unit waterlines. And I have some questions needed to discuss with the author.

Dear reviewer,

Thank you for the reviewing of our manuscript and for your constructive comments.

1. What's the concentration of Bilpron?

Bilpron® is a solution ready for use which was used unmixed. The manufacturer (ALPRO® MEDICAL GmbH) does not provide information on the concentration of the different constituents in the solution (ethylene diamine tetra acetic acid, polyhexamethylene biguanide and ester para-hydroxybenzoate).

2. Is the function of BRS removing the biofilms? What's the difference between ICX and BRS ? Why did you choose the BRS disinfectant? And I have not found the results of Microbiological analysis after the BRS disinfection treatment.

The BRS® is also a product for shock disinfection used to remove the bacteria and biofilms inside DUWLs. The difference with ICX Renew® is its composition and its usage. Like explain in the manuscript, ICX Renew® consists of hydrogen peroxide, sodium lauryl sulfate and maleic acid while BRS® of a 2-phase basic cleaning system including 3 solutions: BRS® PreCleaner (enzymatic cleaning agent containing sodium carbonate and disodium metasilicate), then BRS® Remover (citric acid crystals) mixed with BRS® Activator (surfactants).

We first used ICX Renew® because it is the procedure of the manufacturer to treat the new installed dental chairs. Then, we choose BRS® because our hospital procedures recommend the use of BRS® when a microbiological analysis of DUWLs water is non-compliant (action level) based on a previous study carried out in our hospital [Baudet A, Lizon J, Martrette JM, Camelot F, Florentin A, Clément C. Efficacy of BRS® and Alpron®/Bilpron® Disinfectants for Dental Unit Waterlines: A Six-Year Study. Int J Environ Res Public Health. 2020;17(8):2634. doi:10.3390/ijerph17082634].

The results of microbiological analysis after the BRS® disinfection treatment were written after the Table 3: “After the BRS® disinfection of the three non-compliant DUWLs, all the control samples were compliant without any bacteria detected” (in other words, 0 CFU was detected after BRS® treatment).

3. There were 24 water samples from the water syringes, micromotor and handpiece supply tubes. How many water samples from the water syringes? How many water samples from the micromotor and handpiece supply? Please list.

Water samples (500 ml) were taken simultaneously from the water syringes and from micromotor and handpiece supply tubes. The six DUWLs' outlets of each dental chair (two water syringes, two micromotor supply tubes, one turbine handpiece supply tube and one ultrasonic handpiece supply tube) counted for 1/6 of the total water sample.

4. In the discussion, the second hypothesis and the third hypothesis can not be departed clearly. Because on 25 and 28 November, a leak test of each DUWL was performed by the fitter using the hospital main water supply during the installation. And the initial shock disinfection was performed on

30 November. If the contamination occurred during the installation of the dental chairs, and the microorganisms could also grow and form biofilms inside the DUWLs during 29 to 30 in November. The tube of waterline was already wet because of the leak test. Also no *P. aeruginosa* was tested from the tap water samples, which could not reveal there was no *P. aeruginosa* in tap water. Because *P. aeruginosa* could grow and form biofilms in the tube, so that they could be tested in the dental water samples and not be tested in the tap water samples.

P. aeruginosa was tested in tap water sampled on 21 December (the day where the dental chairs were delivered in our hospital, a few days before their installation). No *P. aeruginosa* was detected from tap water, it is why we rejected this third hypothesis. For a better understanding we added a subsection in the “Experimental procedures” section to describe sampling and analyses of tap water, and we add a paragraph at the end of the results section to present results of these tap water samples.

5. I suggested that the microbiological analysis could be performed before the leak test to analyze whether the contamination origin was from the factory.

We completely agree with you. A future study with microbiological analysis before the leak test would make it possible to analyze whether the contamination origin did not come from the hospital. An installation of 17 new dental chairs is scheduled in our hospital, we will design a new analysis plan to investigate the origin of the contamination.

We added a sentence at the end of the discussion section to highlight future research perspectives.

Reviewer #2

I have a few questions and comments to share with them:

Dear reviewer,

Thank you for the reviewing of our manuscript and for your constructive comments.

Experimental procedures:

1. Disinfection of the Dental Unit Waterlines: why are so many different disinfectants used for disinfection prior to sampling and what is the significance of Alpron® and Bilpron® being used at this stage?

We first used ICX Renew® because it is the procedure of the manufacturer to treat the new installed dental chairs. Then, we choose BRS® because our hospital procedures recommend the use of BRS® when a microbiological analysis of DUWLs water is non-compliant (action level) based on a previous study carried out in our hospital [Baudet A, Lizon J, Martrette JM, Camelot F, Florentin A, Clément C. Efficacy of BRS® and Alpron®/Bilpron® Disinfectants for Dental Unit Waterlines: A Six-Year Study. *Int J Environ Res Public Health*. 2020;17(8):2634. doi:10.3390/ijerph17082634].

Our study was carried out under current practice conditions governed by our hospital procedures. In our hospital, Alpron® and Bilpron® disinfectants are used continuously in DUWLs: Alpron® during periods of activity (Monday to Friday), Bilpron® during inactivity periods (weekends). The ICX Renew® treatment was completed on Friday afternoon (2 December). The laboratory team sampled 12 dental chairs (2nd floor) on Friday, but did not have enough time to sample the 12 dental chairs on the 1st floor. These DUWLs were treated with Bilpron® during the weekend and then sampled the next week.

2. Sampling: Is it sufficient to use 20ml of neutralizer on a 500ml sample to remove the disinfectant? Please indicate whether the timing of the tap water collection of the sample is consistent with the timing of the dental chair waterlines collection.

The 500ml samples were collected from DUWLs fitted by Alpron® diluted at 1% with sterile water. Only 5ml of disinfectant were present in the samples, so 20ml of neutralizer are sufficient.

Tap water collection occurred on 21 December (the day where the dental chairs were delivered in our hospital, a few days before their installation). For a better understanding, we added a subsection in the “Experimental procedures” section to describe sampling and analyses of tap water.

Microbiological analysis:

1. Please describe in detail the process of microbial culture, e.g., what temperature is taken for how many days of culture, etc., which is very unfriendly to the reader, although reference standards have been listed.

As indicated in the “Microbiological analysis” subsection: “The details of the microbiological analysis were described in a previous study [14]”. This reference is an open access article that readers can easily consult to learn about the process of microbial cultures. We do not prefer to copy and paste from this previous paper to do not unnecessarily lengthen the size of our manuscript.

In this previous paper, it was written that: “Total culturable aerobic bacteria counts were performed according to the international standard for water quality: enumeration of culturable microorganisms—colony count by inoculation in a nutrient agar culture medium (NF EN ISO 6222). In brief, two samples of 1mL of water were placed in two sterile 90mm plastic petri dishes, followed by the addition of 20mL of plate count agar (PCA) to each plate, and mixed well. The agar was allowed to harden at room temperature. Thus, one plate was incubated at 36 +/- 2°C and the other was incubated at 22 +/- 2°C for 44 +/- 4 and 68 +/- 4h respectively. The colonies on each plate were counted

immediately after incubation. *Legionella* sp. and *L. pneumophila* detection was carried out according to the international standards for water quality: enumeration of Legionella (NF EN ISO 11731-2). Briefly, 0.2mL of water was inoculated directly to glycine–vancomycin–polymyxin–cyclohexamide (GVPC) agar. Then, 100mL and 10mL of water were separately filtered through a black membrane made of mixed esters of cellulose (pore size 0.45µm). The filters were overlaid with 30mL of pH 2 acid for 5 +/- 0.5min. The filters were rinsed with 30mL of sterile water and placed on GVPC agar plates. The cultures were incubated at 36 +/- 2°C for at least seven days, and CFU were counted at day three, day five, and day seven or more. Suspect colonies were subcultured on buffered-charcoal-yeast-extract (BCYE) agars with and without cysteine for one day. Detection of *P. aeruginosa* was performed according to the international standard for water quality: detection and enumeration of *Pseudomonas aeruginosa*—method by membrane filtration (NF EN ISO 16266). In brief, 100mL of water was filtered through a white membrane made of mixed esters of cellulose (pore size 0.45µm), which was subsequently placed on Cetrimid agar (CN-agar) and incubated at 36 +/- 2°C for 44 +/- 4h. Colonies were counted and examined under UV radiation. The suspect colonies were tested with oxidase reagent and subcultured on King’s B medium. Total coliform bacteria and *E. coli* were researched according to international standard for water quality: detection and enumeration of *Escherichia coli* and coliform bacteria—method by membrane filtration (NF EN ISO 9308-1). Briefly, 100mL of water was filtered through a white membrane made of mixed esters of cellulose (pore size 0.45µm), which was subsequently placed on a Lactose Triphenyl Tetrazolium Chloride (Lactose TTC) agar plate and incubated at 36 +/- 2°C for 44 +/- 4h. Colonies were counted, and suspect colonies were tested with oxidase reagent and subcultured in tryptophan broth tube for indole test.”

2."Each water sample was subjected to analysis for total culturable aerobic bacteria (TCAB) at 22 and 36°C, *Legionella* sp., *Pseudomonas aeruginosa* and total coliform bacteria including *Escherichia coli*". This sentence is a bit vague. Please indicate whether these different culture assays use the same 500ml sample or different samples collected at the same time.

We completed the “Microbiological analysis” subsection to precise how the 500 ml of water sampled from each dental chair were distributed for each analysis (1 ml for TCAB at 22°C, 1 ml for TCAB at 36°C, 0.2, 10 and 100 ml for *Legionella*, 100 ml for *Pseudomonas* and 100 ml for coliforms. We sampled 500 ml to have a margin of safety.

3.What is the significance of using BRS® to sterilize substandard dental chairs in the study ?

As Alpron® and Bilpron®, we used BRS® because our hospital procedures recommend the use of BRS® when a microbiological analysis of DUWLs water is non-compliant (action level). Like explain in our first responses, this procedure is validated by a previous study performed over a 6 years period in our hospital.

We think it is interesting to present the good results obtain with BRS® in this study because it presents a pragmatic solution to dentists and infection prevention and control team to disinfect DUWLs contaminated by bacteria resistant to an initial shock treatment.

We completed the “Corrective measure and control sampling” subsection in the “Experimental procedures” section to explain the choice of BRS®. And we completed our discussion to highlight the importance of this result.

Results:

1.The results of incubation of tap water samples should also be presented in the results section.

We agree with you: the results of tap water analysis should be presented in the results section. We added a paragraph at the end of the results section to present these results.

2. In Table 3, the TCAB results for dental chairs 16 and 17 are all 0, but the results of their *P. aeruginosa* cultures are 4 and 2, respectively. The authors should try to analyze and explain this result in the Discussion section.

We give an explanation regarding these results in our discussion section. The detection of *P. aeruginosa* but not TCAB for dental chairs 16 and 17 can be explained by the difference in volume of water analysed: 1 ml for TCAB at 22°C and 1 ml for TCAB at 36°C compared to 100 ml for *P. aeruginosa*.

Discussion:

The discussion section should provide meaningful insights and offer a comprehensive comparison of the findings with relevant studies in the existing literature. This would enhance the interpretation of the results by providing a broader context and a deeper understanding of their significance. The overall logic of the discussion section is poor.

We have completed our discussion section. But to our knowledge, no previous study has investigated the contamination of DUWLs from new dental chairs, so we cannot compare our results with existing literature. It is why our article is important: it provide new information and it poses hypotheses that will need to be investigated by future research.

Re: Spectrum03962-23R1 (Initial waterline contamination by *Pseudomonas aeruginosa* in newly installed dental chairs)

Dear Dr. Alexandre Baudet:

I have reviewed your responses and revisions; thank you for being careful with those. I have made the decision to accept your manuscript and I am forwarding it to the ASM production staff for publication. Your paper will first be checked to make sure all elements meet the technical requirements. ASM staff will contact you if anything needs to be revised before copyediting and production can begin. Otherwise, you will be notified when your proofs are ready to be viewed.

Sincerely,
Erik Hom
Editor
Microbiology Spectrum